# Deconstructing allostery by computational assessment of the binding determinants of allosteric PTP1B modulators

Adele Hardie[1], Benjamin P. Cossins[2,4], Silvia Lovera[3] & Julien Michel [1✉]

Fragment-based drug discovery is an established methodology for finding hit molecules that can be elaborated into lead compounds. However it is currently challenging to predict whether fragment hits that do not bind to an orthosteric site could be elaborated into allosteric modulators, as in these cases binding does not necessarily translate into a functional effect. We propose a workflow using Markov State Models (MSMs) with steered molecular dynamics (sMD) to assess the allosteric potential of known binders. sMD simulations are employed to sample protein conformational space inaccessible to routine equilibrium MD timescales. Protein conformations sampled by sMD provide starting points for seeded MD simulations, which are combined into MSMs. The methodology is demonstrated on a dataset of protein tyrosine phosphatase 1B ligands. Experimentally confirmed allosteric inhibitors are correctly classified as inhibitors, whereas the deconstructed analogues show reduced inhibitory activity. Analysis of the MSMs provide insights into preferred protein-ligand arrangements that correlate with functional outcomes. The present methodology may find applications for progressing fragments towards lead molecules in FBDD campaigns.

[1] EaStChem School of Chemistry, Joseph Black Building, University of Edinburgh, Edinburgh EH9 3FJ, UK. [2] UCB Pharma, 216 Bath Road, Slough, UK. [3] UCB Pharma, Chemin du Foriest 1, 1420 Braine-l'Alleud, Belgium. [4] Present address: Exscientia, The Schrödinger Building, Oxford Science Park, Oxford, UK. ✉email: julien.michel@ed.ac.uk

The ever-developing experimental methods and automation in the drug discovery industry have led to the popularity of high throughput screening[1] for discovering molecules that bind to proteins of interest, a frequent consideration in the early stages of a drug discovery program. However, the chemical space of drug-like small molecules (Molecular Weight in the range 350–500 g.mol$^{-1}$) is estimated to be around $10^{20}$–$10^{24}$ molecules[2], making it challenging to put together representative screening libraries of a reasonable size. Fragment based drug design (FBDD) is a more efficient strategy that relies on screening initially lower molecular weight compounds (under 300 g.mol$^{-1}$). As the number of unique fragment-sized molecules is much smaller than that of drug-like small molecules, it is easier to routinely screen a greater fraction of fragment chemical space[3]. Structure-enabled biophysical assay methods[4–6] used to screen fragment libraries (MD mix, high-throughput X-ray crystallography) frequently identify numerous fragments that bind all over the surface of the protein of interest. Since fragments are usually weak binders, they only provide starting points for medicinal chemistry efforts to produce a lead molecule active in functional assays[7]. This can be a slow and expensive process, with an uncertain outcome especially in cases where the fragment selected for elaboration binds to a site that is distinct from the active site. In this case it may be difficult to anticipate whether fragment elaboration will lead to a ligand that affects protein function through allosteric mechanisms[8,9].

Molecular dynamics (MD) simulations provide an approach to avoid many of the issues associated with biophysical assays[10,11]. For instance, fragments can be restrained to their binding sites, and they can be grown in silico for further analysis before committing resources to synthesis and assays[12]. There are multiple computational methods available to determine the binding affinity of a fragment[13–16], but there is a need for protocols that aim to assess the allosteric effects of fragment. Computational methods that do so would be useful to prioritise selection of fragments for follow-up elaboration.

Markov State Modelling (MSM) has been used successfully to model the conformational dynamics of proteins[17–22]. An MSM is a transition matrix that describes the conditional probability of a system transitioning to some state $j$ given that it is in state $i$, after a given lag time $\tau$[23–25]. These are then combined to give the distribution of protein conformations, e.g. catalytically active and inactive. The system is treated as memory-less, meaning that the transition probabilities only depend on the current state of the system, and not any previously visited states[24]. A benefit of MSMs is that models of protein dynamics can be built efficiently from multiple short MD trajectories, making use of parallel computing[23]. MSMs, as well as other ensemble methods, have been previously employed to investigate allosteric modulation[26–29].

The data available from standard MD simulations might still be insufficient to explore the complete protein conformation ensemble, such as short-lived intermediate states. Efforts have been invested into adaptive sampling protocols to build more robust MSMs in a unsupervised manner[30–32]. However it is also possible to exploit prior knowledge about key protein conformational states to generate pathways for conformational transitions between them[22,33], using for instance steered MD (sMD) simulations. sMD introduces a restraint on the system, biasing it towards a certain conformation[34,35]. Therefore conformations that are important to protein dynamics but are not sampled easily through equilibrium MD simulations can be accessed.

Here we propose a joint sMD/MSM protocol, illustrated in Fig. 1, to evaluate the effect a ligand has on the conformational ensemble of a protein. In particular, sMD trajectories are not used directly in the MSMs, but only to push the system towards desired intermediate states. Shorter equilibrium simulations, started from conformations achieved via sMD, are employed to observe transitions from these intermediate configurations, which then are combined to build an MSM that uncovers a fuller picture of protein dynamics.

This protocol is tested using protein tyrosine phosphatase 1B (PTP1B) as a benchmark system. PTP1B is a negative regulator of insulin signalling[36]. Inhibition of PTP1B has been proposed as a therapeutic strategy for type II diabetes treatments[37]. The protein enzymatic activity is regulated by the conformations adopted by the WPD loop that sits above the active site. The WPD loop adopts two major conformations - open and closed (Fig. 2a). The closed conformation is catalytically active, as it positions the catalytic residues in range of the substrate[38]. The closing and opening of the loop occurs on multimicrosecond timescales[39]. The charged and highly conserved nature of the PTP1B active site has made it challenging to develop orally bioavailable ligands that act as competitive inhibitors. Therefore allosteric inhibition of PTP1B enzymatic activity is an attractive drug design strategy[40].

Three experimentally characterised inhibitors (**1**-**3**) and a fragment binder (**4**) with unknown functional effect were used to validate the methodology (Fig. 2b). Our protocol successfully identified **1** as a potent allosteric inhibitor[37]. **2**, a fragment obtained by deconstructing inhibitor **1** that only shows very weak activity experimentally was classified as inactive by our protocol. **3**, a covalently bound fragment that weakly inhibit PTP1B shows activity intermediate between **1** and **2** in our protocol[41]. Fragment binder **4** is predicted functionally inactive by our approach. Additionally, we use various protocols to assess the effect of restraining a ligand orientation on the predicted activity levels of the protein. Through comparative analysis of the computed protein conformational ensembles we identify specific protein conformational states that could be used as blueprints for virtual screens of novel PTP1B allosteric modulators. Our efforts illustrate how our joint sMD/MSM protocol could be used to prioritise fragment for hit-to-lead chemistry efforts, and to plan virtual screening campaigns.

## Results

**Validation of the sMD/MSM protocol on substrate simulations.** Systems including *apo* PTP1B, PTP1B with peptide substrate (reference), and PTP1B with substrate and each of the compounds **1**-**4** (Fig. 2), were put through the sMD/MSM workflow as follows. Steered MD simulations were performed, steering the WPD loop and the allosteric network residues, outlined in Supplementary Figs. 1 and 2. From each trajectory, 100 snapshots evenly sampling the observed WPD loop conformations were saved and used as starting points for follow-up 50 ns seeded MD simulations (200 trajectories, 10 μs total sampling time per model). Effects of prolonging the seeded MD simulations to 100 ns are shown in Supplementary Fig. 3. Each trajectory was reduced to two features: WPD loop (residues 178–184) backbone root mean square distance (RMSD) to closed conformation, and the P loop (residues 214–218) backbone RMSD to closed conformation (Fig. 3a). Featurised data of all systems considered here was pooled and clustered into 100 microstates using $k$-means clustering, an example of which is shown in Fig. 3b. Implied timescales (ITS) were computed using a range of lag times between 0.01 and 30 ns (Supplementary Fig. 4), and the final lag time chosen for all MSMs was 20 ns. Chapman–Kolmogorov (CK) tests for each MSM are available in Supplementary Fig. 5.

MSMs were built for all of the systems, using the $k$-means cluster centres as states to transition between. From state transition probabilities, the equilibrium probabilities of each state were computed. When not visited by a particular system, states

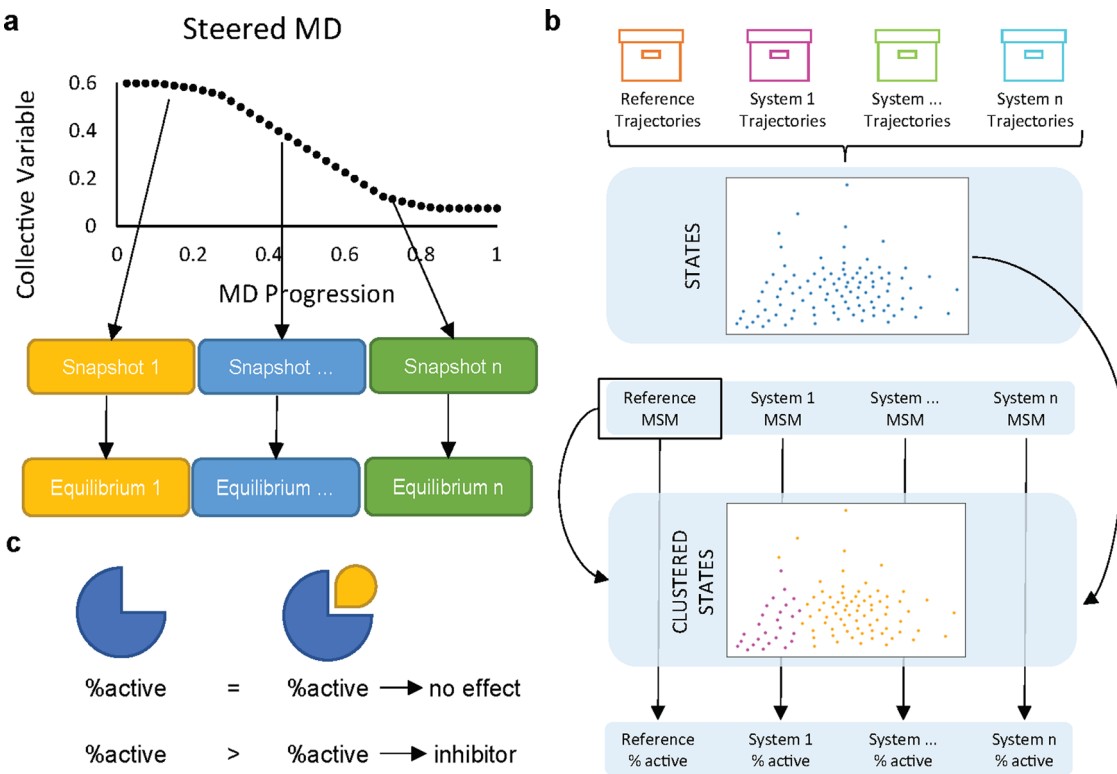

**Fig. 1 The proposed joint sMD/MSM protocol for identifying allosteric modulators. a** Steered MD trajectories are used to generate transitions between protein conformations that are presumed functionally active and inactive for the system of interest. Snapshots from the sMD trajectories are then used as seeds for a swarm of equilibrium MD simulations. **b** The resulting ensemble of MD trajectories for multiple states of the protein of interest are clustered together to generate a consistent definition of microstates for MSM construction. PCCA analysis is used on a reference system to generate a two-state definition to evaluate the percentage of active state population. **c** Comparing the probability that the system will be catalytically active with and without the ligand being investigated allows to model whether it will have an inhibitory effect.

were manually assigned a stationary probability value of 0. Perron Cluster-Cluster Analysis (PCCA) of the reference MSM was used to further cluster the states into two macrostates, referred to as "active" and "inactive" based on the RMSD of the loops[23,25] (Fig. 3b). The metastable state with lower RMSD values corresponds to the active state, as lower RMSD values correspond to higher similarity to the crystal structure of PTP1B with the loop closed. Additionally, the WPD loop RMSD cutoff values for the metastable states correspond to the RMSD value distribution during equilibrium MD simulations of PTP1B with WPD loop closed and open (Supplementary Fig. 2). Active state probabilities obtained from using PCCA assignments based on other system MSMs give very similar results and are shown in Supplementary Fig. 6. The procedure was repeated a hundred times, using the initial micro- and macrostate definitions, to generate probability distributions for observing active states for each system (Fig. 3c).

The major conformation for *apo* PTP1B is the inactive conformation, in agreement with experimental results that suggest a low fraction of active states (2.5%)[42] ("apo" in Fig. 3c). Upon substrate binding there is a significant increase in active conformation probability, in agreement with experimental data[42] ("reference" in Fig. 3c). However while NMR measurements suggest the active conformation dominates PTP1B's conformational ensemble when the enzyme is bound to a substrate (87% population[42]), the MSM indicates the active state is only formed 25% of the time. Experimental data suggests that activation of PTP1B by closure of the WPD loop is coupled with a disorder-to-order transition of helix $\alpha7$. Owing to the difficulties in reliably simulating such large-scale conformational changes the PTP1B model used in the current study is a truncated variant that lacks

helix $\alpha7$. Experimental evidence shows that a mutant PTP1B-$\Delta7$ lacking helix $\alpha7$ is about 40% less active than wild-type[39]. Thus the incomplete activation of PTP1B in presence of a model peptide substrate is fully consistent with experimental observations. The goal of the present protocol is to classify ligands as allosteric effectors by comparison of relative shifts in active state populations, for which trends (relative to the reference system) are sufficient.

**Compound 1 is modelled as an inhibitor, while the deconstructed analogue 2 shows no inhibition**. The active state probability distribution for compound **1** is significantly shifted towards lower values than that observed for the reference system (Fig. 3c, "**1**"), strongly suggesting that compound **1** behaves as an allosteric inhibitor. This behaviour is consistent with an IC$_{50}$ value of **1** ca. 8 μM[37] reported for compound **1**.

Compound **2** is a smaller analogue obtained by truncation of the aryl-sulfonamide moiety of **1** (Fig. 2b). **2** is reported in literature as a very weak inhibitor (IC$_{50}$ ca. 350 μM)[37]. The initial results (Fig. 3c "**2**") did not show a decrease in active conformation probability, but rather a broad up-shifted distribution. Inspection of the MD trajectories used to build the MSM showed that **2** was only weakly bound and had a tendency to escape its binding site on a timescale of several nanoseconds, casting doubts on the reliability of the results obtained by the protocol. A second MSM was built, this time restraining intermolecular distances between **2** and N193 and E276 with weak flat-bottom biasing potentials (See Fig. 4b and methods). These distance restraints were selected to enforce **2** to adopt a

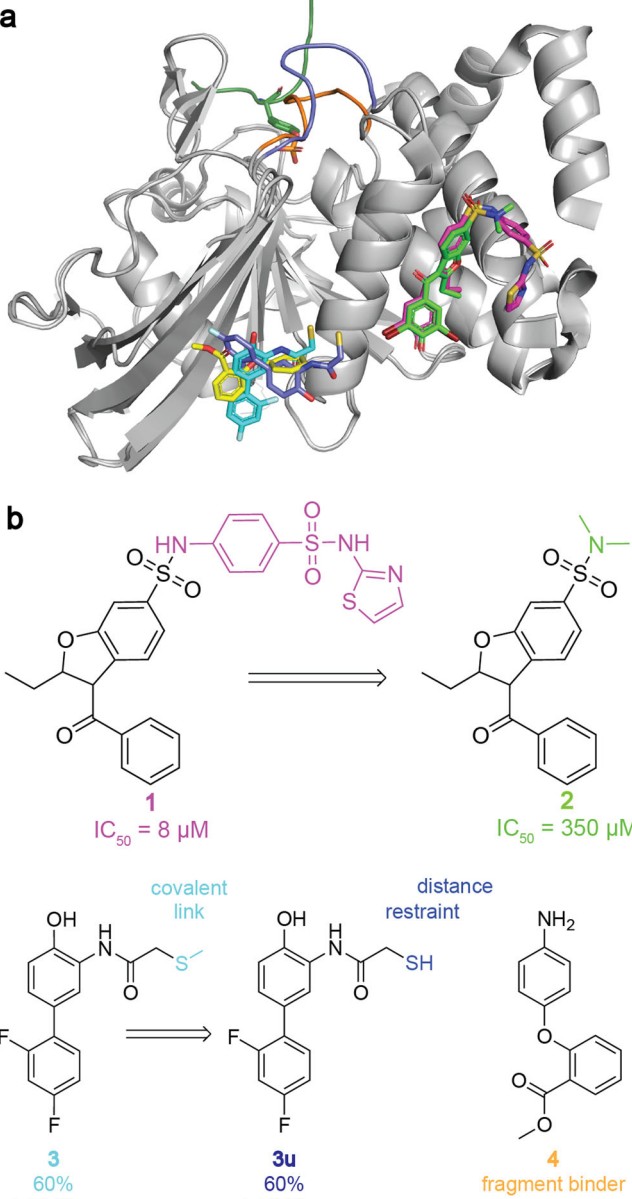

**Fig. 2 Allosteric inhibitors of PTP1B reported in literature. a** PTP1B with WPD loop in active (orange, PDB ID: 1SUG) and inactive (blue, PDB ID: 2HNP) conformations, the substrate peptide (dark green, PDB ID: 1EEO), as well as the allosteric binders shown in **b**: **1** (magenta, PDB ID: 1T4J), **2** (green, PDB ID: 1T48), **3** (cyan, PDB ID: 6B95), and **4** (yellow, PDB ID: 5QDL). Except for "apo" system simulations, all simulations included the peptide substrate as well. **b** Structures of reported allosteric inhibitor **1** and its weak analogue **2**[37]; tethered inhibitor **3**[41] and non-covalently bound variant **3u**; fragment binder **4** with unknown functional effect[41].

binding pose consistent with that observed with **1** throughout the MD simulations. The resulting active state probability distribution (Fig. 3c, "**2r**") was very similar to the reference system, suggesting a lack of functional effect. Extending simulation time or number of simulations reduces model error, but does not suggest inhibitory effect for compound **2r** (Supplementary Fig. 3).

The lack of inhibition by compound **2**, even when restrained to the binding pocket, may relate to its reduced interactions with F280, which has been suggested to be part of the allosteric network of PTP1B[42,43]. Compound **1** wraps around the side chain and π stacks via its thiazole moiety, forcing F280 to adopt primarily an

"up" rotamer (Fig. 4a, d, magenta χ 1 angle ca. −60 deg.). The "up" rotamer of F280 is observed in the inactive sub-ensemble of PTP1B "reference" more frequently than in the active sub-ensemble (Fig. 4c). Compound **2** lacks a arylsulfonamide-thiazole moiety to wrap around F280, and consequently F280 adopts multiple rotameric states during the simulations (Fig. 4b, d green). The most populated "down" rotamer of F280 observed during simulations of **2r** is similar to the major rotamer observed in the active sub-ensemble of PTP1B "reference" simulations (Fig. 4d green χ 1 angle ca. −180 deg. and Fig. 4c, orange). Such differences in behaviour in F280 dynamics are not apparent in crystal structures of **1** and **2** (PDB IDs 1T4J and 1T48) where F280 adopts a "down" rotamer exclusively (Fig. 4d, dashed lines).

**Covalent tethering of compound 3 contributes to allosteric effect.** Large-scale automated crystallography screening of fragments carried out by Keedy et al. has resulted in a tethered fragment **3** at a site distinct from that occupied by compounds **1**-**2**. The fragment is covalently linked to a K197C mutant and shows 60% maximum inhibition[41]. A ligand binding at the K197 site may interact with residues part of the allosteric network, such as Y152 or N193[42,43]. Therefore, the joint sMD/MSM protocol was applied to compound **3**. The model produced a down-shift in active conformation probability distribution with respect to the reference system (Fig. 3c "**3**"), suggesting an inhibitory effect intermediate between **1** and **2**.

In order to further assess the sensitivity of the sMD/MSM workflow to the effect of fragments, a model for untethered **3**, **3u** (with the K197C PTP1B mutant) was built. The covalent linkage was replaced by a flat-bottomed non-directional distance restraint to K197C (see Methods). The sMD/MSM produced a broad active state probability distribution with a median only slightly shifted down with respect to the reference system (Fig. 3c "**3u**"). Comparison of the computed MSM ensembles for **3** and **3u** shows that **3** mainly adopts a "upright" binding pose owing to the covalent tether (Fig. 5a) that resembles the crystallographic pose observed for this fragment. This pose enables the fragment phenol moeity to engage in hydrogen bonding interactions with K150, a suggested allosteric residue[42]. By contrast untethered fragment **3u** is more mobile and adopts predominantly a "sideways" pose (Fig. 5b). This causes the phenol group to interact with E200, which has not been flagged as a residue of interest to the allosteric network[41–43]. The "upright" pose can still be detected albeit less frequently. These observations suggest that stabilisation of the "upright" pose could be a plausible design strategy to elaborate fragment **3** into a non-covalently bound allosteric inhibitor of wild-type PTP1B.

Finally, the protocol was tested on a fragment of unknown allosteric effect. Fragment binder **4**[41] was processed using a similar distance restraint scheme as for **3u**. The resulting active conformation probability distribution for **4** is broad and does not suggest allosteric inhibition when compared with the reference system (Fig. 3c "**4**"). The major binding pose of **4** also corresponds to a "sideway" binding mode that engage in hydrogen bonds with E200, (Fig. 5c) on the α3 helix and adjacent to the binding site of **1** and **2**, but further away from the allosteric residues pictured previously. No minor "upright" pose was detected in the conformational ensemble. Overall these results suggest that fragment **4** does not show potential for allosteric inhibition of PTP1B without further elaboration to enforce adoption of a different binding pose.

**Discussion**
The results reported here demonstrate that the joint sMD/MSM protocol can be used to discriminate allosteric inhibitors from

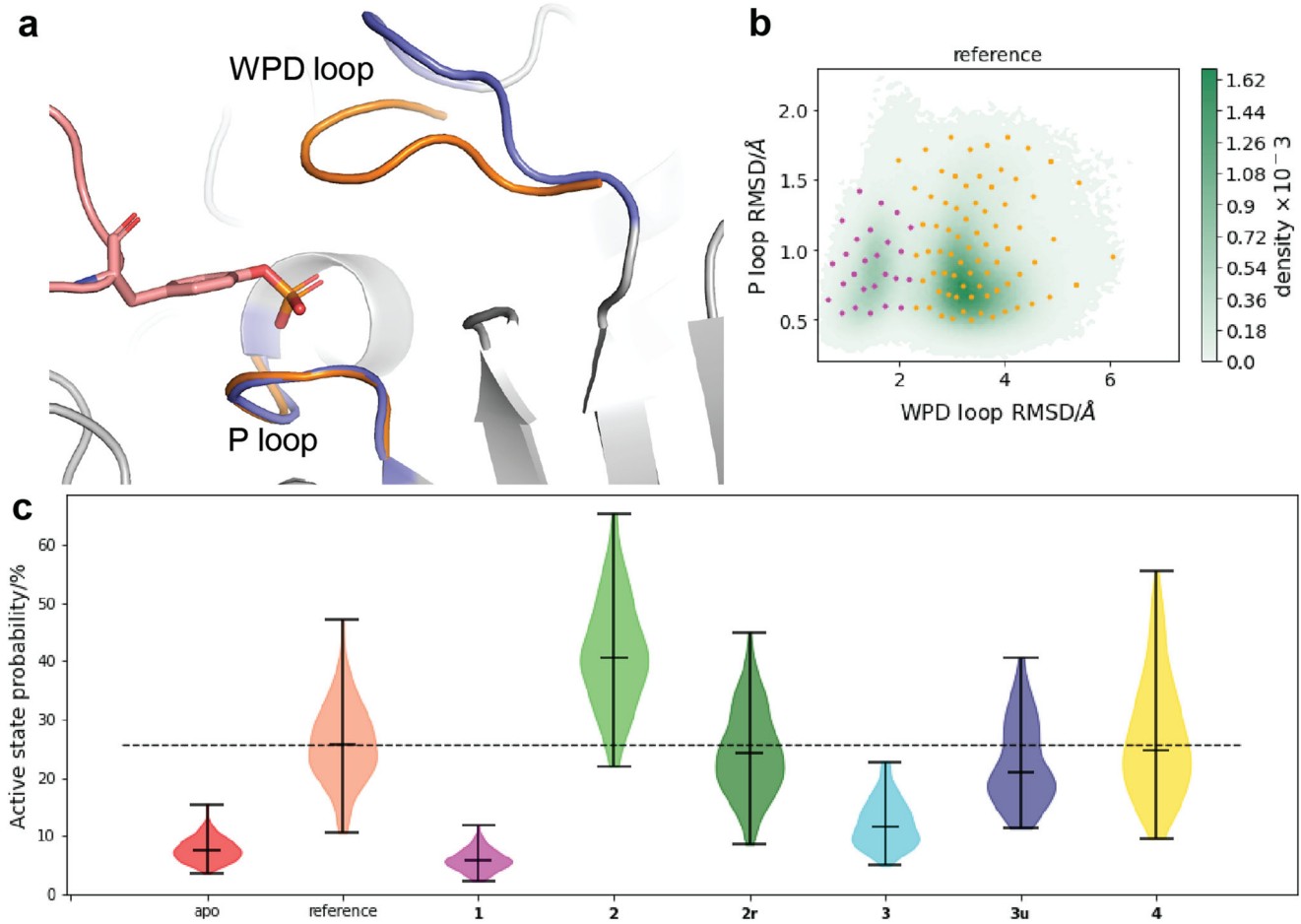

**Fig. 3 Markov State Model features and results. a** The features used to reduce data dimensionality: backbone RMSD to closed WPD loop conformation, and Asp181(Cγ)-Cys215(S) distance. **b** An example of the reference system data, with the microstate clusters overlayed. Each cluster is assigned to a metastable macrostate via PCCA (magenta - active, orange - inactive). Since all of the data was clustered together for consistency, some of the microstates for a given system are not populated. **c** Violin plots of active state probability distributions for each system, after 100 iterations of bootstrapping by resampling. The middle horizontal bar of each violin plot indicates the median active state probability, while the upper and lower bars indicate the maximum and minimum values. The dashed line marks the median active state probability of the reference system. The x axis ticks indicate the PTP1B system composition: *apo* PTP1B (apo), PTP1B with a substrate peptide (reference), and PTP1B with compounds **1-4**, in addition to the substrate peptide (**1-4**). **2r** stands for restrained compound **2**, while **3u** stands for untethered compound **3**, i.e. the covalent S-S bond is replaced with a distance restraint.

non-functional binders. They provide an inverse view of how this workflow could be applied in a computer-aided drug design (CADD) project. The most potent allosteric PTP1B inhibitor reported in the literature (**1**) was analysed and subsequently deconstructed into a less potent variant **2**[37]. The MSM model for **1** suggest potent inhibition in agreement with literature data. Reliable analysis of compound **2** requires the use of restraints to prevent spontaneous unbinding during MD simulations. The judicious use of distance restraints provides information on what interactions are important in the activity of compound **1** and suggests which vectors could be grown or changed to achieve the desired functional results. Similarly, compound **3** is deconstructed into **3u** by replacing a covalent link with an in silico distance restraint, causing a decrease in inhibition. These different strategies to enforce proximity with PTP1B have a significant effect on the conformation of the ligand, and the interactions that are formed with the protein. Compound **4** behaves similarly to **3u**, demonstrating how the protocol may be used to profile compounds with unknown allosteric potential. Further developing **3u** or **4** to behave more like covalently linked **3** (such as moving the compound **4** acetyl group around the benzene ring) could lead to increased efficacy as allosteric inhibitors. Through modelling

active state probabilities via MSMs, these binding pose changes can be related to protein activity.

As the modelled change in active state probability can be related to local changes in ligand binding site conformations, the seeded MD trajectories can be mined to select protein conformations associated with functional states. In turn, the resulting conformations can be used for further virtual screening, to find ligands that could induce the same binding site rearrangements. For example, the increased activity of compound **1** over compound **2** was related to differences in the preferred conformations of F280 during the MD simulations. This insight was not apparent from available X-ray crystallographic data since in existing crystal structures for **1** and **2** this residue is modelled in the "down" conformation (PDB IDs: 1T4J and 1T48 respectively)[37]. The simulations carried out here revealed an alternative "up" rotamer, which is predominantly adopted in inactive states of PTP1B. Therefore targeting the F280 "up" rotamer offers a potential focus for further drug discovery campaigns.

A key feature of the present approach is the use of steered MD simulations. Previous studies applying MSMs to study allosteric modulation have been successful in using unbiased MD

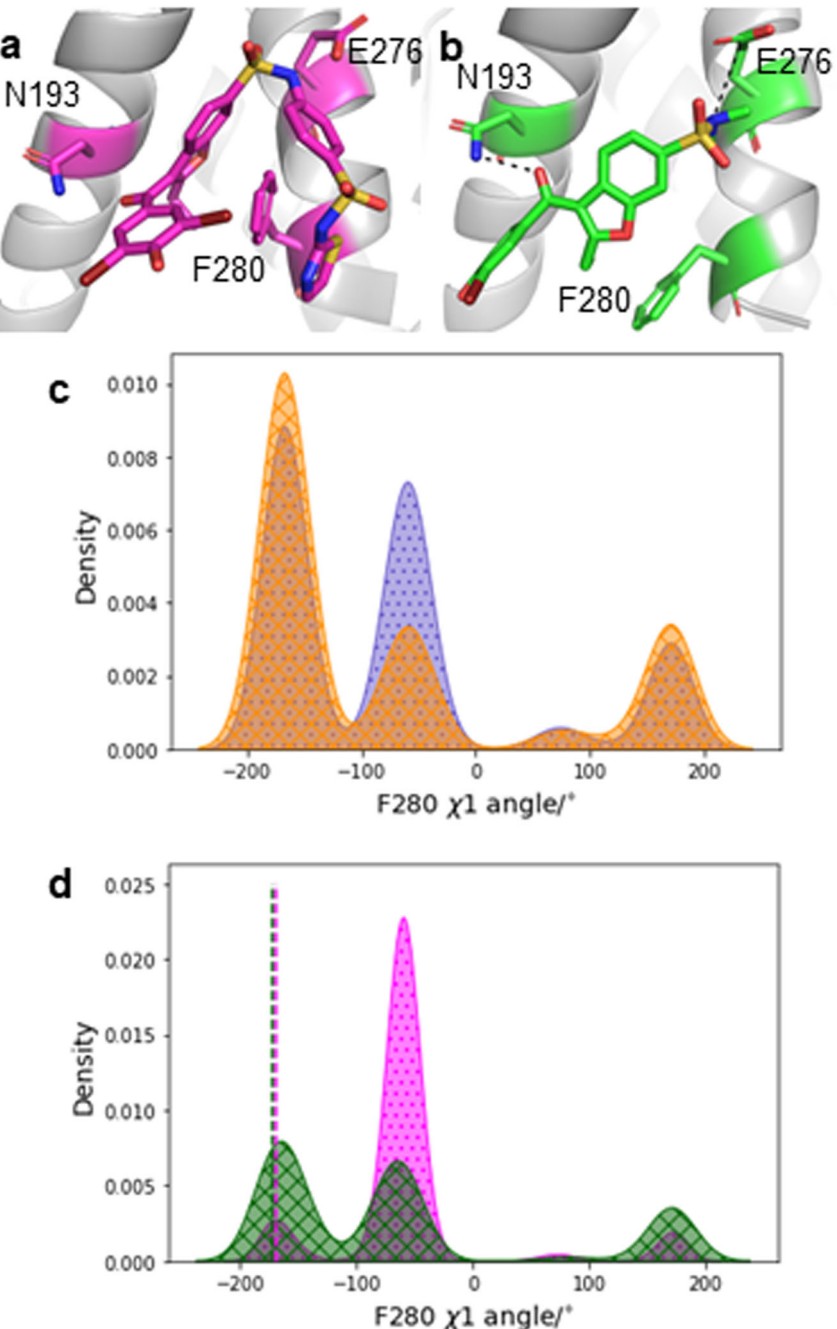

**Fig. 4 Protein and ligand conformations for PTP1B with compounds 1 and 2r. a** Compound **1** and key residues. **b2** and key residues. Distance restraints indicated by black dashed lines. **c** F280 $\chi$1 dihedral when PTP1B is active (orange, crosses) and inactive (blue, dots). **d** F280 $\chi$1 dihedral for PTP1B with **1**(magenta, dots) and **2r**(dark green, crosses). X-Ray values for structures with compounds **1** (PDB ID: 1T4J) and **2** (PDB ID: 1T48) are shown as (overlapping) dashed lines.

simulation data[44]. However, since the WPD loop of PTP1B changes conformation on multi-$\mu$s timescales[39], the simulation length required to observe a number of transitions that is statistically significant is unpractical for routine applications. sMD allows access to intermediate conformations with simulations on nanoseconds timescales, and the following short seeded MD simulations leverage parallel computing, reducing answer time even further. Sampling of relevant conformations, both of the ligand and the protein, is key to modelling activity probabilities consistent with experimental data. Steering only the active site residues does not ensure the allosteric network will adjust to new conformational states as the WPD loop moves, which causes

inconsistent simulation results in disagreement with reported literature values (Supplementary Fig. 7). Future work may focus on extending the enhanced sampling methodologies used to seed the MSMs to decrease the amount of experimental data required to ensure that the relevant protein conformational states have been sampled.

The use of Markov State Modelling enables a decrease of the time-to-answer by modelling long-time scale dynamics as a set of shorter timescale simulations that may be run concurrently. However, obtaining equilibrium distributions require the use of dimensionality reduction. The MD data is reduced to chosen features, which in turn are clustered into discrete microstates.

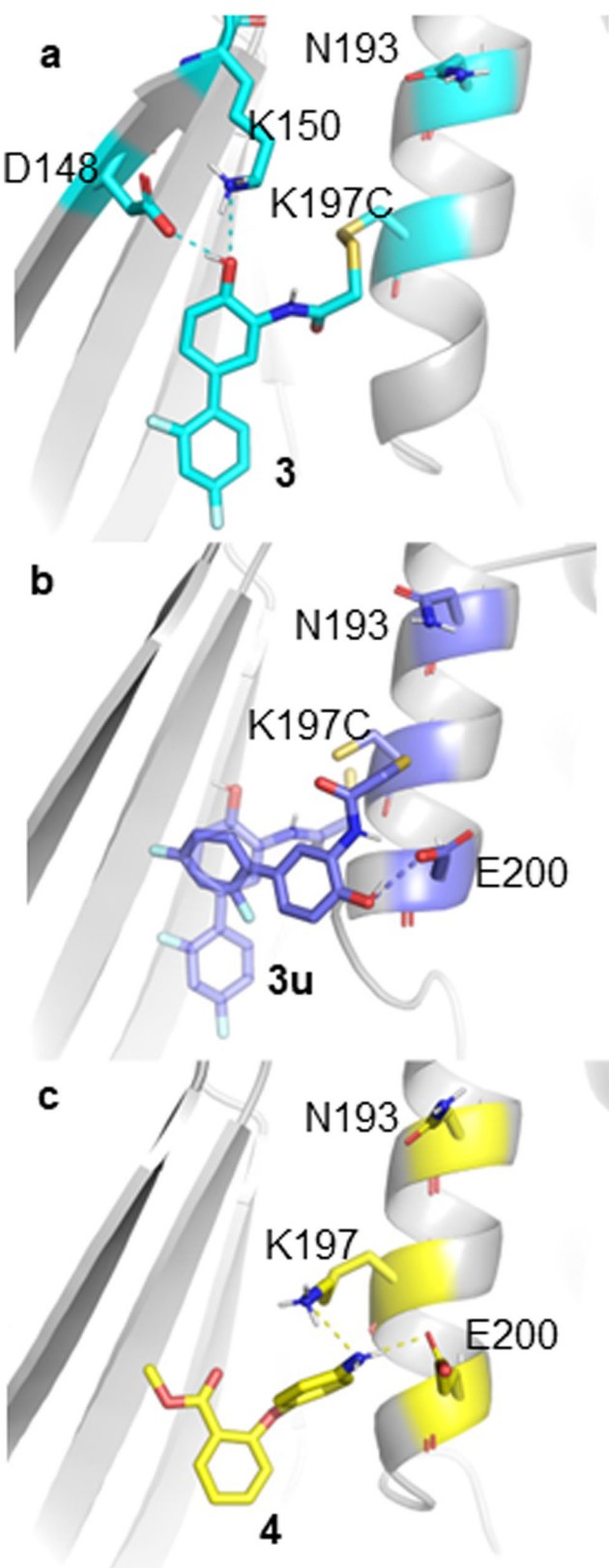

**Fig. 5 The major ligand conformations during seeded MD of 3 (cyan), 3u (dark blue) and 4 (yellow). a** Covalently linked **3** maintains its crystal binding pose, and forms hydrogen bonds with D148 and K150. **b** Replacing the covalent link with a distance restraint changes the binding mode, and interactions are mainly formed with E200 instead of K150. **c** Fragment **4** binds similarly to **3u**.

Those discrete states in this case were assigned to final "active" and "inactive" PTP1B states using Perron Cluster-Cluster Analysis (PCCA), which uses the eigenvectors of the transition matrix that makes up the MSM to find metastable states[45]. Tools that make this procedure simpler and data-driven, such as VAMPnets[31] are in development. The procedure is more complex when comparing multiple MSMs, rather than focusing on a single model. It is preferable that both the microstates and the coarser active/inactive assignments are based on the same feature values for the models to be more easily comparable. Additionally, to determine the active and inactive state partition, the assignments from PCCA of the reference system were used throughout to keep them consistent. However, seven total MSMs were built in this case, and any of those assignments could be used. Supplementary Fig. 6 shows the effects of using different active state definitions on the active state probability and ranking. The results remain qualitatively consistent but in general the automated selection of a suitable macrostate definition is non-trivial. Therefore future work focusing on automating the MSM construction and analysis steps is desirable to facilitate deployment of the technology at scale.

The present study relies on simulation of binders to assess their potential allosteric effects when bound to different pockets on the surface of a protein. Future developments of the protocol could be sought to allow characterisation of the allosteric potential of cryptic binding sites discovered by molecular dynamics simulations, enabling protein druggability assessments before efforts to identify binders are initiated[21,46].

Overall the present results suggest that it is now viable to routinely compare numerous Markov State Models to assess the effects of ligand binding or point mutations on protein function. Extension of the present sMD/MSM methodology to other drug target classes is warranted to validate the generality of the approach for supporting allosteric drug design workflows.

## Methods

**System preparation**. All systems with the WPD loop closed used protein coordinates from PDB ID 1SUG. All open loop protein conformations were from PDB ID 2HNP, with W179 rotated to match the rotamer in 1SUG using Flare v5[47]. Both protein conformations included residues 1–282, truncating the $\alpha$7 helix. Peptide substrate was taken from PDB ID 1EEO. Missing PTP1B residues and all peptide ACE/NME caps were added using Flare.

In all cases, E97 was modelled as GLH and H214 as HID, due to predicted pK$_a$ values of 8.59 and 3.71 respectively by propka3[48], for PDB ID 2HNP. Additionally D181 was modelled as ASH, and C215 as CYM, to match the proton-donor role of D181 and the coordination of substrate by C215. All system preparation was done through BioSimSpace[49], except in the case of tethered ligand **3**. The ff14SB force field was used for protein residues, with additional phosphate parameters from Case et al.[50]. The ligands were parameterised using GAFF2 and the AM1-BCC charge method. Ligand source PDB IDs and charges are outlined in Supplementary Table 1. In all cases, TIP3P water was used to explicitly solvate the system as a cuboid box, with 10 Å distance. Na$^+$ ions were added to neutralise the system, and Na$^+$ and Cl$^-$ ions were added to achieve 150 mM NaCl concentration. All systems were minimised and equilibrated using GROMACS version 2020.2[51]. Minimisation was carried out over 7500 steepest descent steps. Systems were heated from 0 K to 300 K over 100 ps. Equilibration was performed in the NPT ensemble for an additional 250 ps.

**Tethered ligand parameter setup**. To prepare the parameters for covalently linked **3**, the ligand and C197 (atoms SG, CA, CB, HB1-3) residues were obtained from PDB ID 6B95. CA in C197 was changed to a hydrogen, and the CYS was renamed to CYX. The PDB file was converted to mol2 format using antechamber[52] and the AM1-BCC charge method, with a neutral charge. The atom types in the mol2 file were set as follows: SB was set to S, CB to CT and HB1-3 to ha. The force field modification file was generated using parmchk2 and the parameter file was generated using tLeAP. The information corresponding to the Cys197 residue was removed from the parameter file, also modifying connectivity and atom number entries.

**Steered MD**. Steered molecular dynamics were run with Amber20[52] and PLUMED v2.6.1[35] via BioSimSpace. The collective variables used are outlined in

Supplementary Fig. 1. In all cases the first 4 ps were used to apply the force, maintaining the CVs at original values. Open to closed loop conformation sMD was carried out over 150 ns with a 3500 kJ mol$^{-1}$ force constant, while closed to open sMD was carried out over 100 ns with a 2500 kJ mol$^{-1}$ force constant. The target values for the allosteric residues were taken from 1 μs equilibrium MD simulations (Supplementary Fig. 2). All simulations were run at 300 K and 1 atm.

**Seeded MD**. 100 snapshots were extracted from each sMD trajectory (200 total per model), equally sampling the WPD loop RMSD range, using cpptraj v4.25.6 (AmberTools20). The systems were resolved and re-equilibrated as outlined above and 50 ns equilibrium MD simulations were carried out with Amber20 via BioSimSpace, saving snapshots every 10 ps (5000 frames per simulation). Total sampling time of trajectories used for a single MSM was 10 μs.

**Ligand restraints**. In the cases when ligands were restrained for sMD or seeded MD simulations, flat bottomed distance restraints were used. The exact parameters are given in Supplementary Table 2.

**Markov state modelling**. The seeded MD trajectories were featurised using cpptraj[53]. The features used were WPD loop (residues 178–184) backbone RMSD to PTP1B with WPD loop closed (PDB ID 1SUG), and P loop (residues 214–219) RMSD. All MSM model buiding was done using PyEMMA version 2.5.7[54]. All system data was pooled together, and k-means clustering (100 cluster centres) was used to define microstates. Implied timescales were calculated using a range of lag times between 1 and 3000 steps (10 ps to 30 ns). MSMs were generated with a lag time of 2000 steps (20 ns) in all cases. PCCA analysis of the reference system was performed to define two macrostates, assigning the macrostate with lowest RMSD as the active state. The clusters not sampled by the reference system were assigned to the inactive state. When clusters were not sampled by a particular system, they were assigned 0% stationary probability manually.

Bootstrapping by resampling was carried out for 100 iterations for each system. 200 random trajectories were selected, and the MSM was built using the same 100 cluster centres, and the same active state assignment as above. The stationary probabilities of clusters belonging to the active state were summed to give a single active state probability each time.

**Conformational analysis**. For systems with compounds **1** and **2r** (Fig. 2), 10,000 frames were sampled out of all trajectory data, using the MSM stationary probabilities as weights. These trajectories were used to compute the residue behaviour shown in Fig. 4. To obtain the reference active and inactive conformation ensembles, the same was applied to the reference system. However, instead of the stationary probabilities, metastable distributions for the active and inactive states were used.

## Data availability
All input files and featurized trajectory data to reproduce the findings from this study are available on GitHub at michellab/AMMo.

## Code availability
All python scripts and jupyter notebooks to reproduce the findings from this study are available on GitHub at michellab/AMMo.

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

## Acknowledgements

This work has made use of the resources provided by the Edinburgh Compute and Data Facility (ECDF) and JADE2 via computing time awarded by HecBioSim [EPSRC grant no EP/R029407/1]. The authors thank Lester Hedges for the continuous development of BioSimSpace to facilitate production of the sMD/MSM workflow. This work was supported by UCB and EPSRC.

## Author contributions

A.H.: conceptualisation, methodology, software, validation, formal analysis, investigation, writing - original draft, writing - review & editing, visualisation. B.P.C: funding acquisition, supervision. S.L.: supervision, analysis. J.M.: conceptualisation, methodology, resources, formal analysis, writing - review & editing, supervision, project administration, funding acquisition.

## Competing interests

JM is member of the Scientific Advisory Board of Cresset. All other authors declare no competing interests.
