## [Peer Review File · Communications Chemistry]

Reviewers' comments:

Reviewer #1 (Remarks to the Author):

The authors proposed the methodology using Markov state model (MSM) combined with steered molecular dynamics (sMD) simulations for the assessments of the allosteric inhibitors. The assessments of the allosteric inhibitors by the methodology proposed in this study were in good agreement with the experimental results for the PTB1B.

A methodology proposed by the authors seemed to be effective for searching the allosteric inhibitors, however, I have a few comments as following.

One of the important points in this study is the combination of steered MD with the Markov state model. First the steered MD (sMD) for the transformation between open- and close conformations was required before the seeded MD simulations. This must take a long time and requires large computational costs. Isn't the qualitative comparison of the stabilities of the two conformations (open- and close- conformations) enough for finding of the effective allosteric inhibitors? The methodology using of MSM combined with the general MD for the conformation selection for the allosteric ligand binding to protein has already been reported (The Journal of Physical Chemistry B 121.22 (2017): 5509-5514). The only four starting points for the MD simulations were required for the MSM analysis. Is combining with sMD really required for the assessment of the allosteric inhibitor?

Reviewer #2 (Remarks to the Author):

Hardie et al. has proposed a workflow to study the inhibition effect of allosteric modulators, especially fragments of modulators. In brief, enhanced sampling was performed followed by conformation analysis using Markov State Model (MSM), to explore how great the inhibitor could shift conformation ensemble towards the inactive state. In fact, I think it is a quiet classic protocol to study allostery (although different study would take different techniques in MD simulation and conformation analysis), but it is commendable for compiling the protocol into a Python library. Therefore, I think this article could be accepted by Communication Chemistry after answering several questions below.

(1) In the protocol, steered molecular dynamics (sMD) was performed to acquire seed structures, followed by short equilibrium simulation (50 ns for each seed). I wonder whether such short simulations could converge. Several Indexes like RMSD could be shown. Especially, what I concern is whether shortening or prolonging the simulation would affect the ratio of conformation in active state.

(2) Classification of active or inactive states were done by PCCA. However, as a data-driven method, it is possible that in different cases (modulator 1-4), the clustering center locates differently, and in consequence the active and inactive states are defined differently. (And this might explain the weird observations in modulator 2?) I would recommend applying physics-based method, like applying cutoff values of RMSD of loops to discriminate active or inactive state (which could be acquired by PCCA on substrate-bound system). In other words, the definition of inactive/active state could be more precise.

(3) Definition of ticks in Figure 3c could be added to the figure caption.

(4) As has been mentioned, ensembled based method has already been used for study allostery and allosteric modulators. Therefore, I would suggest a few references on this topic:

λ Predicting Protein Dynamics and Allostery Using Multi-Protein Atomic Distance Constraints, *Structure*, 2017, 25, 546.

λ Activation pathway of a G protein-coupled receptor uncovers conformational intermediates as novel targets for allosteric drug design. *Nat. Commun.* 2021, 12, 4721.

λ Delineating the activation mechanism and conformational landscape of a class B G protein-coupled receptor glucagon receptor, *Comput. Struct. Biotechnol. J.*, 2021, 20, 628.

λ Markov state models and molecular dynamics simulations reveal the conformational Transition of the intrinsically disordered hypervariable region of K-Ras4B to the ordered conformation. *J. Chem. Inf. Model.* 2022, 62, 4222.

Reviewer #3 (Remarks to the Author):

The paper described the use of steered molecular dynamics and the Markov State Model to compare the conformational preference of PTP1B when a reference peptide and several small molecules were bound to the protein. The goal was to identify structural features that might distinguish compounds that could allosterically inhibit the protein from those that could not. The methodologies employed were not new and it would have been more convincing to see more experimental support for their simulation results. Nevertheless, the simulations and analyses identified a potentially useful feature that might help to identify novel allosteric inhibitors of PTP1B in the future. In particular, it identified a structure of F280 in the inactive form of the enzyme that had not been observed in experimental structures. Using this structural feature might improve the finding of new allosteric inhibitors by virtual screening.

Minor points:

“The charged and highly conserved nature of the PTP1B active site has made it a challenging to develop orally bioavailable ligands that act as competitive inhibitors.”: an extra “a” appeared before “challenging”.

“TIP3P water was used to explicitly solvate the system as a cuboid box, with 10 distance.”: some words missing near the end.

Prof. Julien Michel
School of Chemistry,
The University of Edinburgh,
Edinburgh, EH9 3FJ, UK

Edinburgh, May 9th 2023

Dear Reviewers,

Pleased find our answers to all comments received.

Reviewers' comments:

Reviewer #1 (Remarks to the Author):

The authors proposed the methodology using Markov state model (MSM) combined with steered molecular dynamics (sMD) simulations for the assessments of the allosteric inhibitors. The assessments of the allosteric inhibitors by the methodology proposed in this study were good agreement with the experimental results for the PTB1B.

A methodology proposed by the authors seemed to be effective for searching the allosteric inhibitors, however, I have a few comments as following.

The authors are thankful to the reviewer for their comment on our work.

One of the important points in this study is the combination of steered MD with the Markov state model. First the steered MD (sMD) for the transformation between open- and close conformations was required before the seeded MD simulations. This must take a long time and requires large computational costs. Isn't the qualitative comparison of the stabilities of the two conformations (open- and close-conformations) enough for finding of the effective allosteric inhibitors? The methodology using of MSM combined with the general MD for the conformation selection for the allosteric ligand binding to protein has already been reported (The Journal of Physical Chemistry B 121.22 (2017): 5509-5514). The only four starting points for the MD simulations were required for the MSM analysis. Is combining with sMD really required for the assessment of the allosteric inhibitor?

We agree that the use sMD simulations introduce more complexity than unbiased MD simulations.. However, transitions between open and closed loop conformations of the WPD loop are rare events (on the micro to millisecond timescale) and in the absence of a sMD run it is difficult to build reproducible MSMs as there is very little overlap in conformations sampled by simulations started from open or closed loop states. The sMD protocol ensures

reproducible transitions between open and closed loop conformations regardless of the ligands complexed to PTP1B. . Additionally, only relatively short MD simulations started from the seeded MD runs are necessary to obtain reproducible results. Therefore using sMD followed by seeded MD can reduce the overall computational resources required. We thus believe that the sMD protocol is a more robust approach able to tackle even more complex conformational changes in other systems well beyond the reach of equilibrium MD simulations.

Qualitatively comparing conformation stability can be subjective, whereas state probabilities modelled by the MSMs are objective and we have shown that they are reproducible. Using the bootstrapping by resampling of the swarm of seeded MD trajectories also gives an estimate in the confidence of the predictions, which informs on the quality and trustworthiness of the model.

We have also made changes in paragraph 3 of the introduction and paragraph 3 of the discussion to further acknowledge general MD ensemble methodology for study of allosteric modulation.

Reviewer #2 (Remarks to the Author):

Hardie et al. has proposed a workflow to study the inhibition effect of allosteric modulators, especially fragments of modulators. In brief, enhanced sampling was performed followed by conformation analysis using Markov State Model (MSM), to explore how great the inhibitor could shift conformation ensemble towards the inactive state. In fact, I think it is a quiet classic protocol to study allostery (although different study would take different techniques in MD simulation and conformation analysis), but it is commendable for compiling the protocol into a Python library. Therefore, I think this article could be accepted by Communication Chemistry after answering several questions below.

Many thanks for the insightful comments of the reviewer.

(1) In the protocol, steered molecular dynamics (sMD) was performed to acquire seed structures, followed by short equilibrium simulation (50 ns for each seed). I wonder whether such short simulations could converge. Several Indexes like RMSD could be shown. Especially, what I concern is whether shortening or prolonging the simulation would affect the ratio of conformation in active state.

We have carried out a number of tests to assess the reproducibility of our protocol, and found that the results were fairly robust. Longer simulation times have no significant effect on the results, but are beneficial to decrease the width of the distribution of active state probabilities.

The following has been added to paragraph 1 of the Results section: “

Different protocols were tested by extending seeded MD duration or adding more seeded trajectories. There was no change in modelled active state probability, but the model error decreased (Supplementary Figure 3).. ”

(2) Classification of active or inactive states were done by PCCA. However, as a data-driven method, it is possible that in different cases (modulator 1-4), the clustering center locates differently, and in consequence the active and inactive states are defined differently. (And this might explain the weird observations in modulator 2?) I would recommend applying physics-

based method, like applying cutoff values of RMSD of loops to discriminate active or inactive state (which could be acquired by PCCA on substrate-bound system). In other words, the definition of inactive/active state could be more precise.

We have tested the robustness of our protocol by systematically varying the state assignments based on PCCA performed on other systems. We find that, while there is some variability, the overall conclusions are not affected. It is also reassuring that the PCCA assignments are consistent with the fluctuations observed in long equilibrium simulations of the WPD loop in closed and open states.

The following has been added to paragraph 2 of the Results section: "The metastable state with lower RMSD values corresponds to the active state, as lower RMSD values correspond to higher similarity to the crystal structure of PTP1B with the loop closed. Additionally, the WPD loop RMSD cutoff values for the metastable states correspond to the RMSD value distribution during equilibrium MD simulations of PTP1B with WPD loop closed and open (Supplementary Figure 2). Active state probabilities obtained from using PCCA assignments based on other system MSMs give very similar conclusions and are shown in Supplementary Figure 6."

(3) *Definition of ticks in Figure 3c could be added to the figure caption.*

Tick definitions have now been added to caption of Figure 3c.

(4) *As has been mentioned, ensemble based method has already been used for study allostery and allosteric modulators. Therefore, I would suggest a few references on this topic: I Predicting Protein Dynamics and Allostery Using Multi-Protein Atomic Distance Constraints, Structure, 2017, 25, 546.*

I Activation pathway of a G protein-coupled receptor uncovers conformational intermediates as novel targets for allosteric drug design. Nat. Commun. 2021, 12, 4721.

I Delineating the activation mechanism and conformational landscape of a class B G protein-coupled receptor glucagon receptor, Comput. Struct. Biotechnol. J., 2021, 20, 628.

I Markov state models and molecular dynamics simulations reveal the conformational Transition of the intrinsically disordered hypervariable region of K-Ras4B to the ordered conformation. J. Chem. Inf. Model. 2022, 62, 4222.

The above references have been included to paragraph 3 of the introduction, acknowledging the previous use of ensemble methods in studying allostery.

Reviewer #3 (Remarks to the Author):

The paper described the use of steered molecular dynamics and the Markov State Model to compare the conformational preference of PTP1B when a reference peptide and several small molecules were bound to the protein. The goal was to identify structural features that might distinguish compounds that could allosterically inhibit the protein from those that could not. The methodologies employed were not new and it would have been more convincing to see more experimental support for their simulation results. Nevertheless, the simulations and

analyses identified a potentially useful feature that might help to identify novel allosteric inhibitors of PTP1B in the future. In particular, it identified a structure of F280 in the inactive form of the enzyme that had not been observed in experimental structures. Using this structural feature might improve the finding of new allosteric inhibitors by virtual screening.

We thank the reviewer for their comments on our work.

Minor points:

“The charged and highly conserved nature of the PTP1B active site has made it a challenging to develop orally bioavailable ligands that act as competitive inhibitors.”: an extra “a” appeared before “challenging”.

Typo corrected.

“TIP3P water was used to explicitly solvate the system as a cuboid box, with 10 distance.”: some words missing near the end.

This was an error in the TeX document not showing “Å”, which has now been corrected.

Best wishes,

REVIEWERS' COMMENTS:

Reviewer #1 (Remarks to the Author):

The manuscript has been improved, and it is in a nice condition now.

Reviewer #2 (Remarks to the Author):

The authors have clarified my concerns, and the current manuscript is suitable for publication.